

# Abundance modulates the ecosystem functional contributions of two sympatric Caribbean sea cucumbers

Rachel Munger, Hannah V. Watkins, Jillian C. Dunic and Isabelle M. Côté

Department of Biological Sciences, Simon Fraser University, Burnaby, British Columbia, Canada

## ABSTRACT

In highly diverse systems such as coral reefs, many species appear to fulfil similar ecological roles, suggesting that they might be ecologically equivalent. However, even if species provide similar functions, the magnitude of those roles could modulate their impact within ecosystems. Here, we compare the functional contributions of two common, co-occurring Caribbean sea cucumber species, *Holothuria mexicana* and *Actynopyga agassizii*, in terms of ammonium provisioning and sediment processing on Bahamian patch reefs. We quantified these functions through empirical measures of ammonium excretion, and in situ observations of sediment processing coupled with fecal pellet collections. On a per-individual level, *H. mexicana* excreted approximately 23% more ammonium and processed approximately 53% more sediment per hour than *A. agassizii*. However, when we combined these species-specific functional rates to species abundances to produce reef-wide estimates, we found that *A. agassizii* contributed more than *H. mexicana* to sediment processing at 57% of reefs (1.9 times more per unit area across all surveyed reefs), and more to ammonium excretion at 83% of reefs (5.6 times more ammonium per unit area across all surveyed reefs), owing to its higher abundance. We conclude that sea cucumber species can differ in the rates at which they deliver per capita ecosystem functions but their ecological impacts at the population level depend on their abundance at a given location.

# INTRODUCTION

Ecosystem functioning is described by the movement of energy and material within an ecosystem, the fluxes of which are controlled by the identity and abundance of species within it (*Loreau, 2000*; *Cardinale et al., 2006*; *Tilman, Isbell & Cowles, 2014*). Trait-based approaches have been used to predict how species influence core ecosystem processes (*Zwart, Solomon & Jones, 2015*; *Cadotte, 2017*; *Carturan, Parrott & Pither, 2018*), which on coral reefs include herbivore–algae interactions, predator–prey interactions, nutrient cycling, and carbonate dynamics (*Brandl et al., 2019*). These approaches have been useful when applied to highly biodiverse ecosystems where an ecological function is fulfilled by multiple species that share one or more characteristics relevant to that function (*Strong et al., 2015*; *Brandl et al., 2019*; *Wolfe et al., 2020*). Traits such as body mass, trophic group,

Corresponding author
Isabelle M. Côté, imcote@sfu.ca

and nutrient stoichiometrics are widely used as proxies for functional richness to predict the role of groups of species on ecosystem functioning (*McGill et al., 2006*; *Bellwood et al., 2019*). However, the use of trait-dependent groups or guilds can mask significant interspecific variation, often oversimplifying species contributions to ecological processes (*Semmler et al., 2021*). For instance, parrotfishes contribute to the process of carbonate dynamics through coral grazing and bioerosion, but species differences in body size can lead to disproportionate contributions to bioerosion by large species at the reef-wide scale (*Lange et al., 2020*). For example, one species of bioeroding parrotfish was found to remove 27 times more material per year than another co-occurring and closely related species (*Bellwood, 1995*). There is therefore value in empirically measuring individual-level functional rates to infer the importance of species to specific ecosystem processes, even within functional guilds (*Lange et al., 2020*).

The role of vertebrates, especially fishes, has dominated studies of ecosystem functioning on coral reefs (*Allgeier et al., 2014*; *Allgeier, Burkepile & Layman, 2017*), even though invertebrates are dominant contributors to coral reef ecosystem diversity (*Glynn & Enochs, 2011*). Specifically, mobile invertebrates only make up a small proportion (7.3%) of the literature on species' functional niches (*Bellwood et al., 2019*) yet can make up a substantial proportion of biomass and play a range of ecological roles. For example, sea cucumbers (Echinodermata: Holothuroidea) are found at high densities (up to three individuals per m$^2$) in seagrass beds and near reef flats (*e.g.*, *Lee et al., 2018*). They serve functional roles in nutrient recycling, sediment processing, and benthic primary productivity in tropical systems (*Uthicke & Klumpp, 1998*; *Uthicke, 1999*; *Uthicke, 2001a*; *Wolkenhauer et al., 2010*; *MacTavish et al., 2012*; *Purcell et al., 2016*). As detritivores that feed on epibenthic organic material and meiofauna, holothuroids can turnover 64–250 kg of sediment individual$^{-1}$ yr$^{-1}$ (*Wolfe & Byrne, 2017*; *Hammond, Meyers & Purcell, 2020*; *Williamson et al., 2021*), with some of this activity occurring nocturnally (*Hammond, 1982*; *Navarro et al., 2013*). Sediments egested by sea cucumbers are often lower in organic matter than the sediment consumed (*Mercier, Battaglene & Hamel, 1999*), demonstrating the role deposit-feeding sea cucumbers can play in re-mineralization of surface sediments (*MacTavish et al., 2012*). Some sea cucumber species can cause changes in sediment grain size, potentially through calcium carbonate dissolution and sediment abrasion during digestive processes (*Hammond, 1981*; *Schneider et al., 2011*). Additionally, sea cucumbers excrete inorganic nitrogen as ammonium ($NH_4^+$), providing a nitrogenous source for benthic microalgae (*Mukai et al., 1989*; *Uthicke & Klumpp, 1998*; *Uthicke, 2001a*), macroalgae (*Felaco, Olvera-Novoa & Robledo, 2020*) and seagrass (*Wolkenhauer et al., 2010*; *Arnull et al., 2021*). For example, in the Indo-Pacfic, field enclosures with high densities of the sea cucumber *Holothuria scabra* resulted in a 30% increase in seagrass leaf extension rate compared to low-density enclosures (*Arnull et al., 2021*). Owing to their high densities in some habitats, sea cucumber populations have the potential to contribute a consistent 'press' of ammonium within marine systems (*Allgeier, Burkepile & Layman, 2017*). This is especially important on oligotrophic coral reefs, which primarily rely on nutrient recycling to bolster benthic primary productivity (*Hatcher, 1988*; *Allgeier, Burkepile & Layman, 2017*). Taken together, the functional roles provided by sea cucumbers can form a link in transferring

energy to other marine trophic levels through the functions of nutrient recycling and sediment processing (*Purcell et al., 2016*; *Wolfe et al., 2020*).

In this study, we investigated the nutrient provisioning and sediment processing functions of two common sea cucumber species in a shallow coral reef–seagrass ecosystem in The Bahamas. Our research objectives were to use empirical measures of ammonium excretion rates and field observations of sediment processing rates to estimate and compare per capita and reef-wide contributions by *Holothuria mexicana* and *Actinopyga agassizii* to both ecosystem processes. Since *H. mexicana* is exploited disproportionately more in the Caribbean than *A. agassizii* (*Rogers et al., 2018*), determining the magnitude of nutrient contribution and sediment processing by these species to coral reefs will provide insight into the potential functional loss associated with current and future sea cucumber fisheries.

## MATERIALS & METHODS

### Study location and study species

Research was conducted under a Marine Scientific Research Permit issued by the Department of Marine Resources, Government of The Bahamas, to the Cape Eleuthera Institute, and in accordance with the Canadian Council of Animal Care (Protocol No. 1301B-19).

The study was conducted on 35 separate coral reef patches along the southwestern coast of Eleuthera Island, The Bahamas, from May to August 2019 (Fig. 1A). Reef patches were located in Rock Sound, a large, shallow (<5 m depth) sandy basin. They ranged in hard-bottom area from 2 to 209 $m^2$ (mean $\pm$ sd: 35 $\pm$ 43 $m^2$) and depth from 2.6 to 4.5 m (mean $\pm$ sd: 3 $\pm$ 0.5 m) and were separated from the nearest patch by a minimum of 100 m. All patch reefs were immediately surrounded by a halo of seagrass, *Thalassia testudinum*, that extended up to 9.6 m away from the patch edge (Fig. 1B). Beyond this distance, seagrass was either sparse or absent. The two focal sea cucumber species, *H. mexicana* and *A. agassizii* (family Holothuriidae; Fig. 1C), are distributed widely across the Caribbean region (*Hendler et al., 1995*). In Rock Sound, we found both species co-occurring in seagrass beds and on or near coral patches. *Holothuria mexicana* feeds approximately 12 h per day, whereas *A. agassizii* feeds approximately 10 h each day (*Hammond, 1982*). There was no sea cucumber fishery in The Bahamas at the time of this study. A small experimental sea cucumber fishery in the Bahamas took place in 2010 on Andros Island but closed after 11 months due to stock depletion (*Sherman et al., 2018*).

### Seagrass area, sea cucumber body sizes and density

We estimated seagrass area at each patch by measuring the circumference at the outer edge of the high-density seagrass halo (Fig. 1B), as well as that of the hard-bottom area of the coral patch reef itself. We converted perimeters to areas and subtracted the hard-bottom area from the total area to obtain seagrass area. Divers counted, identified to species, and measured the length and midbody girth of every sea cucumber encountered on reefs and within the dense seagrass halo. Beginning at a recognizable landmark and moving in a clockwise fashion, two divers swam side-by-side, and systematically searched in the seagrass for sea cucumbers, and then searched the reef, carefully looking in crevices and overhangs

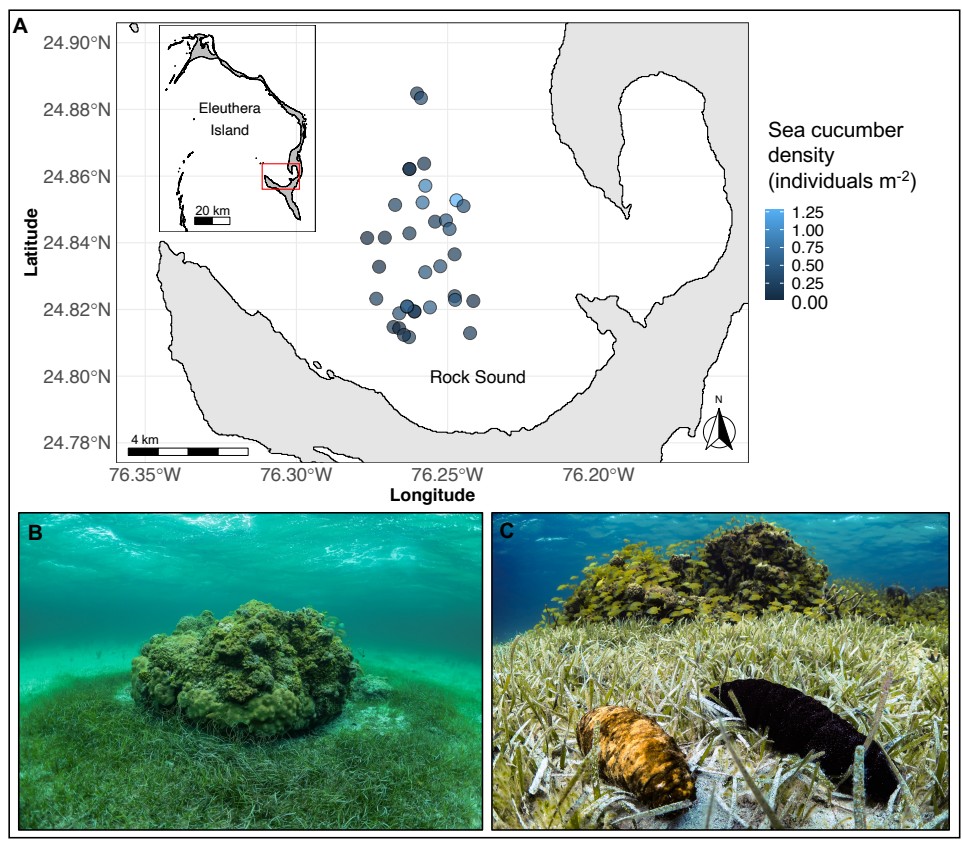

**Figure 1** **(A) Map of Eleuthera Island (inset), The Bahamas, and the study area, Rock Sound (main panel), (B) one of 35 patch reefs surveyed in this study, and (C) co-occurring *Actinopyga agassizii* (left) and *Holothuria mexicana* (right).** In (A), each dot is a patch reef, whose colour represents the total density of sea cucumbers at each patch, within the surrounding halo of dense seagrass (size of dots not drawn to scale).

for sea cucumbers. Sea cucumber species density is expressed as individuals per m² of seagrass area.

## Sediment processing and movement

We quantified hourly sediment processing by *A. agassizii* and *H. mexicana* following *Lee et al.* (*2018*, adapted from *Uthicke, 1999*), where we assumed that the quantity of sediment egested is equal to the quantity ingested. We selected *A. agassizii* (mean length ± sd = 20 ± 2 cm, range: 14–23 cm, n = 20) and *H. mexicana* (mean length ± sd = 24 ± 3 cm, range: 18–30 cm, n = 20) individuals that represented the commonest size classes across Rock Sound. We measured sediment processing and movement for 20 individuals of each species from 11:00 to 16:00 hrs in July 2019 on two patch reefs that were separate from our 35 survey patches. To do so, divers tracked sea cucumbers by planting a metal stake labelled with flagging tape in the sediment at a standardized distance (ca. 1 cm) from the posterior end of the focal sea cucumber. At the end of each hour (for three consecutive hours), the number of fecal pellets egested by each individual was counted, and the linear distance

moved by each individual was recorded. The stake was re-placed near the posterior end of the focal individual to serve as starting point for the next hour-long observation. After the last observation period, up to 10 (average = 9 ± 0.4 pellets; median = 10 pellets; range = 0–10 pellets) of the most recently defecated fecal pellets for each individual (*i.e.,* the pellets closest to the individual) were collected in Falcon® tubes. More recently released pellets were chosen because they are easier to collect as they have not yet disintegrated. The length and girth of each sea cucumber were also measured. Fecal pellets were frozen and transported to Simon Fraser University.

After thawing and combining pellets for each individual, we placed them in a drying oven for 24 h at 60 °C. Pellets were then weighed to determine dry weight (DW) on an analytical balance to the nearest 0.001 g. Dried pellet samples were transferred into porcelain crucibles, placed in a muffle furnace for 2 h at 550 °C, then reweighed to obtain ash weight (AW). We calculated ash-free dry weight (AFDW = DW –AW) to determine the organic matter (OM) content in the fecal pellets.

### Empirical estimates of ammonium excretion

To measure excretion rates of *A. agassizii* and *H. mexicana*, divers collected individuals of both species haphazardly from various reef patches that were separate from our sediment processing and movement observations. We made excretion estimates by following well-established methods by *Layman et al. (2011)* and *Francis & Côté (2018)* who modified slightly the methods of *Schaus et al. (1997)*, *Whiles et al. (2009)*, and *Taylor et al. (2007)*. Individual *H. mexicana* and *A. agassizii* ($n = 20$ for each species) were brought to the Cape Eleuthera Institute (CEI, 24°49′54.46″N, 76°19′56.28″W) and allowed to recover in sea tables connected to a flow-through seawater system pumped directly from the ocean for 1–2 h before being placed gently but rapidly in individual 20-L acid-washed bags filled with a known volume of pre-filtered (0.7 µm Whatman GF-F filters) sea water. Bags containing sea cucumbers ($n = 20$ per species) and control bags of filtered sea water containing no sea cucumbers ($n$ = three empty bags) were sealed and placed in sea tables to maintain ambient temperature (29–31 °C) for 60 min. Although handling might have increased excretion rate initially, the relatively long incubation period makes it likely that sea cucumbers were near resting rates for most of this period. At the end of the incubation period, we collected one 100 ml water sample from each bag using a sterile plastic syringe. Samples were filtered (0.45 µm Whatman GF-F filters), placed in dark bottles and refrigerated for immediate analysis of ammonium ($NH_4^+$) content, a proxy for inorganic nitrogen, using fluorometric methods (*Taylor et al., 2007*). After incubation, we measured the wet weight (g), total length (cm), and midbody girth (cm) of each sea cucumber and allowed them to recover in sea tables for several hours before release onto their reef of capture. We randomized with respect to species the order in which we measured ammonium excretion of individuals.

### Data analyses

We transformed sea cucumber counts at each patch into densities (*i.e.,* numbers per m² of seagrass). We ran a Welch's *t*-test to assess differences in mean density and mean

proportion of total density between sea cucumber species across Rock Sound. To test for differences in size distributions between *A. agassizii* and *H. mexicana*, we ran a two-sample Kolmogorov–Smirnov test on sea cucumber length.

### Individual-level estimates

To test for species differences in sediment processing, we ran t-tests to examine the effect of species on four metrics related to sediment processing: (1) fecal pellet egestion rate (pellets $h^{-1}$), (2) weight per pellet (g), (3) sediment processing rate (g of sediment $h^{-1}$), and (4) organic matter (% OM). In calculating sediment processing rate, we assumed that the quantity of sediment egested is equal to the quantity of sediment ingested (*Uthicke, 1999*). Additionally, we ran a $t$-test to test for differences in speed (m $h^{-1}$) between *A. agassizii* and *H. mexicana*.

To provide a longer-term perspective on the contributions of individual sea cucumbers and increase comparability with other studies, we extrapolated individual hourly sediment processing rates to individual annual rates. To do so, we converted individual hourly rates of sediment processing from g $h^{-1}$ to kg $yr^{-1}$, assuming sediment processing rates remain constant throughout the year. Following *Lee et al. (2018)*, we multiplied the egestion rate (pellets $h^{-1}$) of each species by 12 h and 10 h of activity for *H. mexicana* and *A. agassizii*, respectively (*Hammond, 1982*) to obtain a daily rate of fecal pellet egestion (pellets $d^{-1}$). We then multiplied this rate by the average pellet weight for each species (g $pellet^{-1}$), which gave sediment weight processed per day (g $d^{-1}$). Finally, we extrapolated this rate to an annual rate in kg of sediment processed $y^{-1}$ per individual.

To test for species differences in hourly ammonium excretion rate ($\mu$mol $NH_4^+$ $h^{-1}$), we ran a linear model with sea cucumber wet weight (data-centred), species and their interaction as model predictors. We used wet weight as a predictor in our model so it could be compared to other studies (see Discussion).

### Reef-level estimates

We used the individual-level estimates of sediment processing (g $h^{-1}$) and ammonium excretion ($\mu$mol $NH_4^+$ $h^{-1}$) described above to generate reef-level estimates of sediment processing (kg $m^{-2}$ $yr^{-1}$) and ammonium excretion ($\mu$mol $NH_4^+$ $m^{-2}$ $h^{-1}$) for each species on each of the 35 reefs by bootstrapping confidence intervals (following *Fieberg, Vitense & Johnson, 2020*).

To generate annual reef-level estimates of sediment processing (kg $m^{-2}$ $yr^{-1}$) for each species, we first used a $t$-test to test the effect of species on sediment processing rate (g $h^{-1}$). To incorporate uncertainty around this relationship, we bootstrapped reef-specific 'populations' of sediment processing rates for 5,000 iterations to obtain reef-level estimates at each reef for each species, accounting for the number of individuals of each species at each site. We then converted our bootstrapped estimates from g $h^{-1}$ to kg $yr^{-1}$. Finally, we divided the bootstrapped estimates of sediment processing by seagrass area to obtain annual sediment processing rate per unit area of seagrass (kg $m^{-2}$ $yr^{-1}$).

We used the same method to estimate reef-level estimates of ammonium contribution ($\mu$mol $NH_4^+$ $m^{-2}$ $h^{-1}$) by each species on each of the 35 reefs. However, because we could record length but not wet weight of sea cucumbers during underwater surveys, we first

converted all observed sea cucumber total lengths of both species to wet weight, using the significant relationship generated between these two variables from the sea cucumbers used to estimate ammonium excretion rates (linear model; $F_{1,36} = 15.56$, $p < 0.001$ for both species). This allowed us to predict ammonium excretion ($\mu$mol h$^{-1}$) as a function of wet weight for each species. To incorporate uncertainty around this relationship, we bootstrapped each reef-specific 'population' of wet weights (5,000 iterations) to predict total ammonium excretion rates for each species at each patch reef, using the model from our empirical excretion measurements. Lastly, we divided our bootstrapped estimates of total ammonium excretion by the seagrass area at each reef to obtain excretion estimates per unit area of seagrass ($\mu$mol NH$_4^+$ m$^{-2}$ h$^{-1}$). All statistical analyses were conducted in R (*R Core Team, 2020*, version 1.3.959) using the tidyverse (*Wickham et al., 2019*), ggspatial (*Dunnington, 2020*), dunn.test (*Dinno, 2017*), cowplot (*Wilke, 2019*), and viridis (*Garnier, 2018*) packages.

## RESULTS

### Body size and density

Across all patches surveyed, *H. mexicana* was significantly larger than *A. agassizii* (mean $\pm$ CI (range); *Hm*: 27 cm [25.9, 27.2] (17–47 cm), *Aa*: 22 cm [21.5, 22.3] (13–45 cm), $p < 0.001$). The length distributions of the two species were also significantly different ($p < 0.001$; Fig. 2). There were, on average, 15 ($\pm$ 2 SE) sea cucumbers per patch (range: 1–62; Fig. 1). There were significantly more *A. agassizii* (11 $\pm$ 2) present, on average, than *H. mexicana* (4 $\pm$ 1) per patch ($p = 0.003$). Mean patch size was 35 ($\pm$ 7 SE; range 1.3–209) m$^2$ and mean seagrass area was 54 ($\pm$ 10 SE; range 5.3–296) m$^2$, yielding a mean overall density of 0.3 ($\pm$ 0 SE; range 0.02–1.3) sea cucumbers per m$^2$ of seagrass area.

### Sediment processing and movement

*Actinopyga agassizii* egested fecal pellets at approximately four times the rate of *H. mexicana* ($p = 0.008$; Fig. 3A). However, the fecal pellets egested by *H. mexicana* were seven times heavier than those egested by *A. agassizii* ($p < 0.001$; Fig. 3B). Combining these measures together, individual *H. mexicana* processed three times more reef sediment per hour, on average, than individual *A. agassizii* ($p < 0.001$; Fig. 3C). Fecal pellets egested by *A. agassizii* had a significantly higher OM by approximately 1.5% than those of *H. mexicana* ($p < 0.001$; Fig. 3D). *Holothuria mexicana* moved a maximum of 170 cm in a three-hour observation, while *A. agassizii* moved at most 125 cm. Speed did not differ significantly between *A. agassizii* (0.1 $\pm$ 0.03 m h$^{-1}$) and *H. mexican* a (0.2 $\pm$ 0.04 m h$^{-1}$) ($p = 0.17$; Fig. 3E).

Through extrapolation of egestion rates and quantities, we found that individual *A. agassizii* and *H. mexicana* have the potential to process 5.9 (range: 4.3–7.5) and 12.5 (range: 9.4–16) kg of sediment y$^{-1}$, respectively. When we scaled up these individual egestion rates to population-level rates, *A. agassizii* populations turned over, on average, significantly more sediment (1.9 $\pm$ 0.2 SE kg m$^{-2}$ yr$^{-1}$) than *H. mexicana* (1.0 $\pm$ 0.4 SE kg m$^{-2}$ yr$^{-1}$; $p = 0.019$; Fig. 4). However, this difference appears due mainly to the absence of *H. mexicana* from several reefs. When we considered only reefs where both species were present (24 of 35 reef patches), *A. agassizii* populations turned over sediment at a similar rate (1.6 $\pm$ 0.2

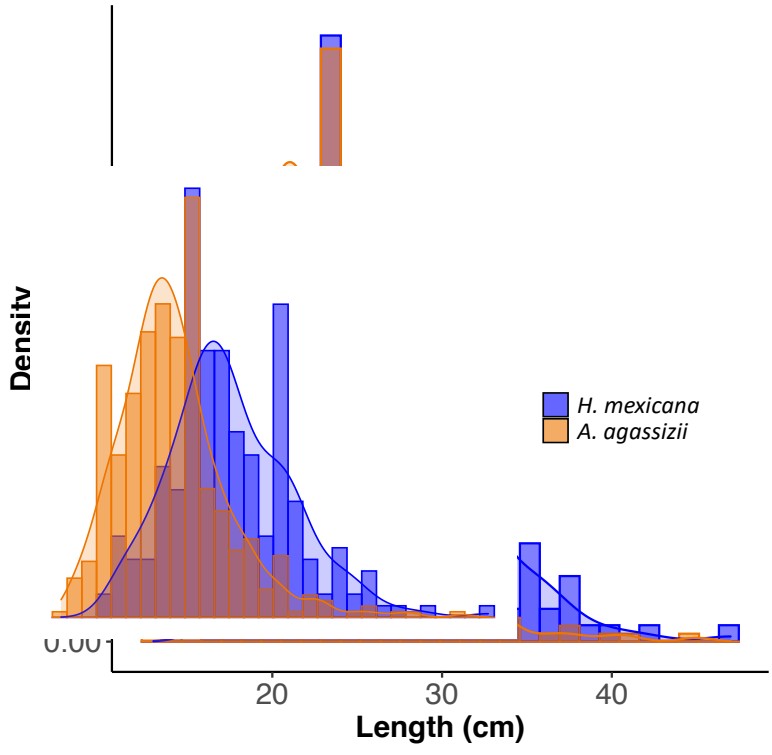

**Figure 2** Size distribution of length for *H. mexicana* (26.5 ± 0.33 cm long, range: 17–47 cm, *n* = 223) and *A. agassizii* (21.9 ± 0.20 cm long, range: 13–45 cm, *n* = 462) from 35 reef patches in Eleuthera, The Bahamas. Density is the proportion of data points in each length bin, which sums to 1.

SE kg m$^{-2}$ yr$^{-1}$) compared to *H. mexicana* populations (1.4 ± 0.5 SE kg m$^{-2}$ yr$^{-1}$; *p* = 0.77; Fig. 4). At reefs with both species, *Actinopyga agassizii* contributed more to sediment processing at 57% of reefs, *H. mexicana* contributed more at 40% of these reefs, and both contributed equally at a single reef (Fig. S1).

## Ammonium excretion

The sea cucumbers used to assess ammonium excretion rates were slightly larger, on average, than those found on the study reefs, but they spanned the ranges of lengths observed on reefs (Fig. 2). We used slightly (~11%) larger *H. mexicana* individuals (mean ± SE [range]; 787 ± 69 [361–1397] g) on average, than *A. agassizii* individuals (mean ± SE [range]; 706 ± 55 [110–1080] g) to obtain ammonium excretion rates. Species identity had a significant effect on sea cucumber ammonium excretion rate (*p* = 0.04), but there was no effect of wet weight (*p* = 0.07, *r*$^2$ = 0.17) (Fig. S2). On average, individual *Holothuria mexicana* excreted NH$_4^+$ at a rate that was approximately 23% higher than individual *A. agassizii* (mean ± SE; *Hm*: 15.6 ± 1.1 µmol NH$_4^+$ h$^{-1}$, *Aa*: 12.0 ± 1.0 µmol NH$_4^+$ h$^{-1}$) (*p* = 0.023).

Reef-level estimates of excretion contributions showed that *A. agassizii* populations contributed 5.7 times more ammonium per unit area (3.1 ± 0.5 µmol NH$_4^+$ m$^{-2}$ h$^{-1}$) than *H. mexicana* populations (0.54 ± 0.1 µmol NH$_4^+$ m$^{-2}$ h$^{-1}$; *p* < 0.001; Fig. 4). When
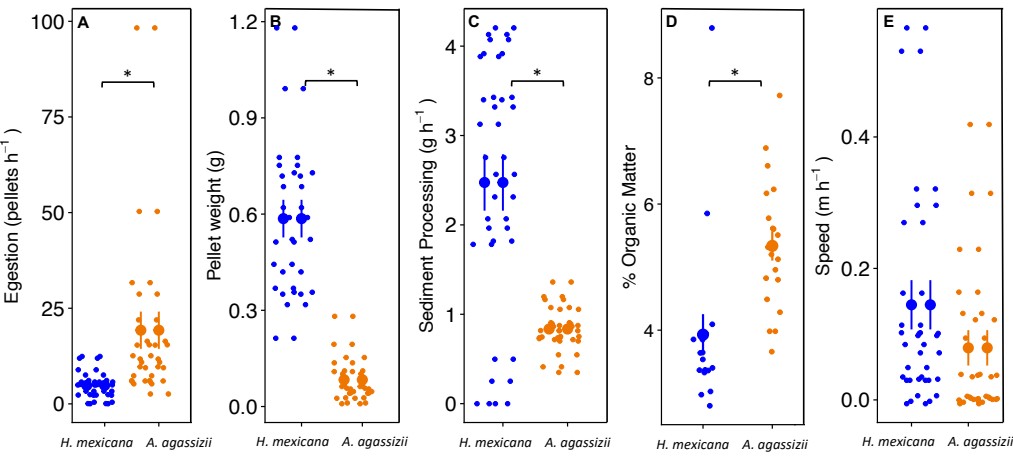

**Figure 3** Dot and whisker plots of (A) fecal pellet egestion rate (number h$^{-1}$), and (B) weight per fecal pellet, (C) sediment processing rate (g h$^{-1}$), (D) % organic matter content of egested fecal pellets, and (E) movement speed of *H. mexican*. Dots display the mean, and whiskers display the 95% confidence intervals. Asterisks denote significant differences between species.

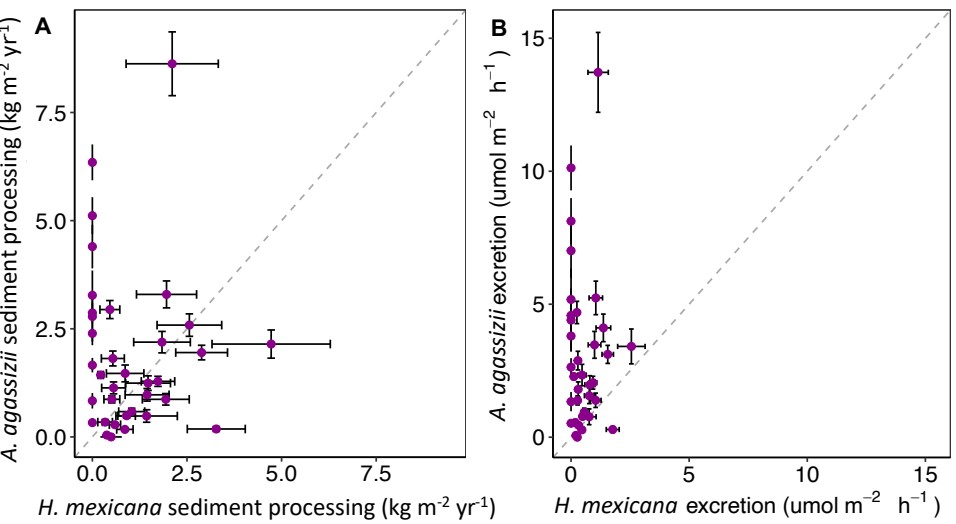

**Figure 4** Estimates of (A) annual sediment processing rate (kg m$^{-2}$yr$^{-1}$) and (B) hourly ammonium excretion rate (mol NH$_4^+$ m$^{-2}$ h$^{-1}$) of *H. mexicana* and *A. agassizii* populations across 35 pat. Each point represents a patch reef. Horizontal and vertical error bars represent the standard error (standard deviation of 5,000 bootstrap resamples) of contributions by *H. mexicana* and *A. agassizii* populations at each site, respectively. The dashed line represents equal contributions by both species.

considering only reefs where both species occurred, *A. agassizii* populations contributed 3.3 times more ammonium per unit area (2.5 ± 0.6 μmol NH$_4^+$ m$^{-2}$ h$^{-1}$) than *H. mexicana* populations (0.75 ± 0.1 μmol NH$_4^+$ m$^{-2}$ h$^{-1}$; $p < 0.001$; Fig. 4). When both species were

present, *A. agassizii* contributed more ammonium at 83% of reefs, *H. mexicana* contributed more at 13% of reefs, and both species contributed equally at one reef (Fig. S3).

## DISCUSSION

This is the first study to evaluate the functional roles of *H. mexicana* and *A. agassizii* in relation to two important ecological processes. Rates of sediment processing and ammonium excretion for *H. mexicana* and *A. agassizii* were species-specific, providing evidence that the two species provide these functions at different magnitudes. Although individual *H. mexicana* processed approximately 53% more sediment and excreted approximately 23% more ammonium per hour than individual *A. agassizii*, differences in abundance between the two species resulted in larger reef-wide contributions to both functions by *A. agassizii* across sites. This suggests that, in the coral reef–seagrass ecosystem we studied, differences in population density can reverse the individual-level differences in the magnitude of the functions provided by these two species of sea cucumbers. Elsewhere, sea cucumber density could simply attenuate or even exacerbate differences in individual rates, depending on both species-specific contributions and abundance.

### Sediment processing and movement

We highlight the ecological roles of *H. mexicana* and *A. agassizii* as motile sediment processors of patch reef sediments. At the individual level, *A. agassizii* egested pellets faster than *H. mexicana*, but their pellets were smaller. Assuming the total amount of sediment egested reflects the amount ingested (*Uthicke, 1999*; *Lee et al., 2018*), individual *H. mexicana* processed three times more sediment per hour than individual *A. agassizii*. The sediment processing rates of both species were within the range of Mediterranean species (*Coulon & Jangoux, 1993*) but considerably lower than those of Indo-Pacific species (*Uthicke, 1999*; *Wolfe & Byrne, 2017*; *Lee et al., 2018*).

At the reef scale, however, *A. agassizii* had a higher estimated sediment processing potential because of its higher abundance. *Actinopyga agassizii* populations processed 1.9 times more sediment per unit area than *H. mexicana* populations. Note that our estimates of sediment processing rates across patch reefs are likely conservative because we could only measure sediment processing rates of sea cucumbers between 09:00 and 16:00 hrs but both species feed and are active at night (*Hammond, 1982*). There is no information, to our knowledge, on the ecological consequences of sediment processing by our two target species. However, the ingestion and release of fecal casts and disturbance caused by locomotion by other deposit-feeding sea cucumbers play a role in redistributing surface sediments and influencing biotic interactions occurring at the sediment–water interface (*Purcell et al., 2016*). Moreover, halting bioturbation and feeding functions by experimental removal of sea cucumbers led to the development of cyanobacterial mats and a reduction in oxygen penetration depth into sediments (*Moriarty et al., 1985*; *Uthicke, 1999*; *Michio et al., 2003*; *Lee et al., 2018*).

*Actinopyga agassizii* egested fecal pellets with higher OM content than *H. mexicana*. This might make pellets of *A. agassizii* more prone to bacterial and fungal growth, and result in a more rapid loosening of the mucous membrane that holds the fecal material

and hence faster resuspension of organic matter (*Conde, Diaz & Sambrano, 1991*). The interspecific difference may stem from differences in the time each species spends on various substratum types. For example, in Jamaica, *A. agassizii* was observed mainly on algal turf and on macroalgae, whereas *H. mexicana* spent 90% of its time on sand (*Hammond, 1982*). Alternatively, interspecific differences may be attributed to the resident time of sediment in the digestive tract, where one species could be less 'thorough' in sediment processing than another, resulting in fecal pellets with higher organic matter. The fecal pellets of *A. agassizii* could also be higher in OM because they are smaller, meaning that there is more organic material surrounding the pellet relative to the amount of sediment inside the pellet, assuming that organic matter is concentrated on the surface of the pellet where most digestive reactions take place. Future studies should (1) use a stable isotope approach to identify the specific origins and diet sources of *H. mexicana* and *A. agassizii* (*e.g.*, *Slater, Carton & Jeffs, 2010*), (2) determine pellet grain size distribution between species, which may have an effect on organic matter content and could reveal niche differentiation, and (3) quantify nocturnal sediment processing (and ammonium excretion) rates of sea cucumbers.

## Ammonium excretion

Individual ammonium excretion rates by *H. mexicana* and *A. agassizii* were species specific but did not significantly vary with body size. The average rates estimated here for *A. agassizii* (12.0 $\mu$mol $NH_4^+h^{-1}$) and *H. mexicana* (15.6 $\mu$mol $NH_4^+h^{-1}$) are at the high end of the range reported for Western Pacific tropical species (1–18 $\mu$M; *Mukai et al., 1989*; *Uthicke, 2001a*; *Wheeling, Verde & Nestler, 2007*). Though *H. mexicana* had a higher average excretion rate, both species showed the same weak relationship with body size, indicating that individual *H. mexicana* excrete more nutrients than individual *A. agassizii* of the same size. Note that the relationship between ammonium excretion rate and sea cucumber body size was weaker than expected from physiology and mass–balance theory. Obtaining accurate but non-destructive mass and morphology measurements of holothuroids is notoriously difficult because they readily change shape and retain water in their body cavity (*Wheeling, Verde & Nestler, 2007*).

Despite having a lower per capita excretion rate, *A. agassizii* contributed more ammonium than *H. mexicana* at the reef scale owing to its higher abundance. *Actinopyga agassizii* contributed more to ammonium excretion at 83% of our study reefs and, on average, excreted 5.6 times more ammonium per unit area than *H. mexicana*. Ammonium excretion by tropical sea cucumbers has been shown to be an important source of limiting nutrients that promotes growth of microalgae (*e.g.*, *Uthicke, 2001b*; *MacTavish et al., 2012*) and seagrass (*e.g.*, *Wolkenhauer et al., 2010*).

Both sea cucumber species together contributed approximately 15% of the ammonium released by coral reef fishes on patch reefs in Rock Sound. We estimated that *A. agassizii* and *H. mexicana* excreted 3.1 $\pm$ 0.5 $\mu$mol $NH_4^+m^{-2}h^{-1}$ and 0.5 $\pm$ 0.1 $\mu$mol $NH_4^+m^{-2}h^{-1}$, respectively. In Rock Sound, all resident fishes together contribute, on average, $\sim$25 $\mu$mol m$^{-2}$h$^{-1}$ during the daytime (*Francis & Côté, 2018*). These fishes included more than 45 species across 17 families. On a per-species basis, the role of sea cucumbers as

nutrient providers is therefore substantial, and given their nocturnal behaviour, they may contribute more to the overall diel nutrient budget at night compared to daytime. In addition, migratory grunts (Haemulidae), which contribute more than twice as other reef-associated fishes, migrate seasonally and annually, resulting in an unpredictable nutrient supply (*Francis & Côté, 2018*). In contrast, some species of sea cucumbers are known to exhibit high site fidelity for years over time (*Wolfe & Byrne, 2017*), meaning that sea cucumbers may contribute more consistently to seagrass beds adjacent to reefs than reef fish do. In this way, sea cucumbers act as a 'press' of nutrient inputs, operating on time scales of days to months, or even years (*Allgeier et al., 2014*; *Allgeier, Burkepile & Layman, 2017*).

## CONCLUSIONS

The two sea cucumber species we studied differed in the per-capita rates at which they deliver two ecosystem functions, but their ecological impacts at the population level depended on their abundance. We draw two main insights from these findings. First, our results are likely place-specific. We examined only two of the many functional roles of sea cucumbers; however, it should be expected that other functions, such as modulation of alkalinity and provision of habitat for symbionts (*Purcell et al., 2016*), will also depend on abundance. This means that the relative importance of co-occurring sea cucumber species in fulfilling these various functions will vary spatially and reflect local patterns of relative species abundance. Second, our results suggest that activities (*e.g.*, fishing) or events (*e.g.*, disease epidemics) that could reduce overall densities will have substantial impacts on the ecological functions provided by sea cucumbers. Moreover, the impacts will be exacerbated if these disturbances affect primarily the species that provide ecological functions at higher per-capita rates. This is the case for sea cucumbers in the Caribbean region, where *H. mexicana* has already been heavily exploited (*Rogers et al., 2018*). In Belize, *H. mexicana* makes up between 65–90% of total sea cucumber catch, while *A. agassizii* makes up approximately 1% (*Rogers et al., 2018*). Identifying the species and populations that contribute disproportionately to ecosystem processes is increasingly important in a time when ecosystems are being transformed and diversity is being lost across ecosystems (*Loreau et al., 2001*).

## ACKNOWLEDGEMENTS

We thank the Cape Eleuthera Institute staff for facilitating field logistics, and Ryan Gateman for field assistance.

### Funding

Rachel Munger was supported by a Graduate International Travel Award from Simon Fraser University, and fieldwork was supported by an NSERC Discovery Grant to IMC (RGPIN-03933-2017). The funders had no role in study design, data collection and analysis, decision to publish, or preparation of the manuscript.

## Grant Disclosures

The following grant information was disclosed by the authors:

Graduate International Travel Award from Simon Fraser University.

NSERC Discovery Grant to IMC: RGPIN-03933-2017.

## Competing Interests

The authors declare there are no competing interests.

## Author Contributions

- Rachel Munger conceived and designed the experiments, performed the experiments, analyzed the data, prepared figures and/or tables, authored or reviewed drafts of the article, and approved the final draft.
- Hannah V. Watkins performed the experiments, analyzed the data, authored or reviewed drafts of the article, and approved the final draft.
- Jillian C. Dunic analyzed the data, authored or reviewed drafts of the article, and approved the final draft.
- Isabelle M. Côté conceived and designed the experiments, authored or reviewed drafts of the article, and approved the final draft.

## Ethics

The following information was supplied relating to ethical approvals (i.e., approving body and any reference numbers):

Research was conducted under a Marine Scientific Research Permit issued by the Department of Marine Resources, Government of The Bahamas, to the Cape Eleuthera Institute, and in accordance with the Canadian Council of Animal Care (Protocol No. 1301B-19).

## Data Availability

The data is available at Zenodo: Rachel Munger. (2022). rachmung/ms_ch1: v1.0 (v1.0). Zenodo. https://doi.org/10.5281/zenodo.7431443.

## Supplemental Information

Supplemental information for this article can be found online at http://dx.doi.org/10.7717/peerj.14823#supplemental-information.

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
