# Peer review of "Abundance modulates the ecosystem functional contributions of two sympatric Caribbean sea cucumbers"

_PeerJ, doi:10.7717/peerj.14823_

## Round 0.1 · original submission · Minor Revisions

I am sorry for the delay on this decision, but the final referee is now a month overdue, so I have decided to move forward without them. Both of the other referees have consistent and positive feedback on your manuscript in terms of the value of the study and the contribution to the field. However, each also points out a number of minor issues that should be addressed or would greatly enhance the value of the data to future readers. I tend to agree that the referees are offering constructive feedback and I encourage you to incorporate it into your revision.

During your revision, please ensure that all review comments (including those on the annotated manuscript) are addressed in a rebuttal letter that outlines how you have addressed each comment. Any edits or clarifications mentioned in the rebuttal letter should also be inserted into the revised manuscript where appropriate. It is a common mistake by authors to address reviewer questions in the rebuttal letter but not make the changes in the revised manuscript. If a reviewer raised a question, then your readers will probably have the same question so you should ensure that the manuscript can stand alone without the rebuttal letter. Directions on how to prepare a rebuttal letter can be found at: https://peerj.com/benefits/academic-rebuttal-letters/ if you need additional guidance.

I look forward to seeing your revised manuscript.

·

Basic reporting

The article meets the PeerJ requirements of basic reporting. The manuscript is well written in a clear, intelligible way. It contains a good introduction into the subject which lays out the rationale for the study. The manuscript draws on the relevant literature and embeds the findings well within the wider body of knowledge.
The structure of the article follows the format suggested by PeerJ. The figures are relevant to the content of the article and are mostly appropriately described and labeled (see below for minor suggestions).
The submission contains most of the necessary information to assess and replicate the methods, and to understand the reported results (see below for minor comments on the data). All results relevant to the objectives of the paper are included.
The raw data and codes used for data analyses are supplied with the manuscript. As there are many supplied files, each with different data labels, a file with meta data and description of the types of data and the column labels would have been helpful for the reader to understand the structure of the data. Also, some of the files contain information that is not understandable without further explanation; for example the files “individual_cukes.csv” and “pred_weights2.csv” contain similar data, but how (and whether) each of them was used is not entirely clear. Also, they contain a column entitled “remove_add_keep” which indicates some kind of treatment of individuals (e.g. don’t consider for analysis, move among patches), but this is not explained anywhere. A description of the data as well as potential manipulation and the rationale would remove ambiguity and aid in replication and appraisal of the presented information and methodology.

Experimental design

The paper constitutes original primary research and is within the aims and scope of PeerJ. The experimental design seems sound and allows for replication of the study (aided by further provision of information on the available data; see first point). The approach is well-described and adequate to address the objectives of the paper. The methods draw on established approaches drawn from the relevant literature. There are some minor points were the data should be presented in a different way to make them more meaningful – specific suggestions are included in the general comments below. All necessary permits, including an ethics review/animal care protocol, are listed.

Validity of the findings

The findings and conclusions are convincing. The data and results are robust, and the conclusions are appropriate and related to the objectives of the paper, including a good consideration of potential limitations and further considerations. A few minor suggestions on improved reporting of results and placement in a wider context (fisheries impacts) are given below.

Additional comments

I found the article interesting and very well-written, with a straightforward study design and reporting of findings. I have the following minor suggestions to improve the manuscript:
Abstract
Lines 34 and 35: for the statements in brackets (“times more per unit area”) – does this refer to the overall average across all surveyed reefs, or only across those reefs where A. agassizii contributed more than H. mexicana?
Introduction
Lines 44-45: Suggest to provide a few references for the first part of this sentence, which deals with a different topic (trait-based approaches) than the second part (ecosystem processes on coral reefs).
Lines 45-46: Should be “herbivore-algae interactions”
Line 65: Suggest to modify to “of the literature on species’ functional niches”
Lines 85-87: italicize species names
Materials & Methods
Line 136: Instead of the maximum value (“up to ten”), it would be more meaningful to provide average, together with the range and median
Results
Lines 235-236: To make the results more comparable and better understandable without knowledge of patch size, suggest to give density per area instead. If presenting density per patch, you should also provide information on (mean) patch size.
Line 255: I am wondering about the approach of the fishery – is there any fishery impacting the study area? This is not entirely clear and could be specified in the “Study location” section. Looking at the size distributions of both species, it does not seem that larger individuals of H. mexicana are removed preferably, and it thus seems fishers may collect all individuals found at a patch reef? Thus, it may be meaningful to also calculate a density of H. mexicana for only those patch reefs where that species was present, instead of across all patch reefs? All assuming that there is fishing in the area.
Discussion
Line 317: This was also observed by Lee et al., 2018 (which you cite elsewhere)
Lines 325-327: This assumes that the organic matter is concentrated on the outside of the pellet - is there evidence of that? I would assume that the digestion while passing through the gut takes place mostly on the surface of the forming pellets - another explanation may be that A. agassizii is less thorough in its digestion of organic matter due to the higher throughput of sediment and shorter time of sediment passing through the digestive tract.
Lines 378-380: Is there any information on the fishery in The Bahamas? Are the species fished there? This should actually be mentioned in the introduction (or description of study location), but could be elaborated further here.
Figure 2, legend: Suggest to call the “Density distribution of length” "Size distribution" instead for ease of understanding. The meaning of “Density” on the y-axis is then still explained in the subsequent sentence, which is sufficient.

·

Basic reporting

the article complies with the basic reporting guidelines for the journal, however, it could have stronger literature dealing specifically with ecological roles of sea cucumbers in nutrient cycling

Experimental design

the experimental design is adequate

Validity of the findings

the findings are valid, but a deeper discussion is recommended, especially regarding the sampling times and sediment sizes in the feces for each species.

Additional comments

it is an adequate paper, could be improved with minor revisions and adding more references, specifically on the effects of sea cucumber excretions on nutrients and its interactions with other organisms, as well as their feeding habits (generally nocturnal) and grain size preferences.

---

## Round 0.2 · accepted · Accept

Thank you for your detailed response to the referee feedback and your thorough revision of the manuscript. I am satisfied that you have addressed all the concerns raised by the referees and the revisions have clarified areas of concern or confusion from the initial submission.

Thus, I am happy to move your manuscript forward into production.
Congratulations and thank you for selecting PeerJ as the outlet for your research!